# Genome-Wide Transcriptomic Identification and Functional Insight of Lily *WRKY* Genes Responding to *Botrytis* Fungal Disease

**DOI:** 10.3390/plants10040776

**Published:** 2021-04-15

**Authors:** Shipra Kumari, Bashistha Kumar Kanth, Ju young Ahn, Jong Hwa Kim, Geung-Joo Lee

**Affiliations:** 1Department of Horticulture, Chungnam National University, Daejeon 34134, Korea; shiprakumari11@yahoo.com (S.K.); bashistha_kanth@yahoo.com (B.K.K.); wnduds357@naver.com (J.y.A.); 2Department of Smart Agriculture Systems, Chungnam National University, Daejeon 34134, Korea; 3Department of Horticulture, Kangwon National University, Chuncheon 24341, Gangwon-do, Korea; jonghwa@kangwon.ac.kr

**Keywords:** Botrytis disease, *Lilium longiflorum*, qRT-PCR, transcription factor, *WRKY family*

## Abstract

Genome-wide transcriptome analysis using RNA-Seq of *Lilium longiflorum* revealed valuable genes responding to biotic stresses. WRKY transcription factors are regulatory proteins playing essential roles in defense processes under environmental stresses, causing considerable losses in flower quality and production. Thirty-eight *WRKY* genes were identified from the transcriptomic profile from lily genotypes, exhibiting leaf blight caused by *Botrytis elliptica*. Lily WRKYs have a highly conserved motif, WRKYGQK, with a common variant, WRKYGKK. Phylogeny of *LlWRKY*s with homologous genes from other representative plant species classified them into three groups- I, II, and III consisting of seven, 22, and nine genes, respectively. Base on functional annotation, 22 *LlWRKY* genes were associated with biotic stress, nine with abiotic stress, and seven with others. Sixteen unique *LlWRKY* were studied to investigate responses to stress conditions using gene expression under biotic and abiotic stress treatments. Five genes—*LlWRKY3*, *LlWRKY4*, *LlWRKY5*, *LlWRKY10*, and *LlWRKY12*—were substantially upregulated, proving to be biotic stress-responsive genes in vivo and in vitro conditions. Moreover, the expression patterns of *LlWRKY* genes varied in response to drought, heat, cold, and different developmental stages or tissues. Overall, our study provides structural and molecular insights into *LlWRKY* genes for use in the genetic engineering in *Lilium* against *Botrytis* disease.

## 1. Introduction

*Lilium* spp. is one of the most important ornamental plants used for cut or potting flowers and gardening purposes. The current international trade of ornamental plants including Lilium is £60–75 billion and increases every year world-wide by 2–4% [1]. However, the lily is subject to a wide range of abiotic and biotic stresses, which have been worsened by global climate changes. *Botrytis* in *Lilium* is a common fungal disease affecting leaves, stems, and sometimes the flowers. The disease is dispersed by spores when splashed around by rain or watering. *Botrytis* affects firstly leaves, flower then bulb until the plant is affected three years consecutively [2]. Low temperature and high humidity cause necrotic lesions on lily leaves and petals, initiating rotting caused by *Botrytis elliptica* [3]. *Fusarium oxysporum* causes basal rot disease, developing black or brownish necrotic lesions on the bulb, resulting in bulb disintegration [4]. Thus, the quality and yield of *Lilium* are affected by various abiotic and biotic stresses. The drought stress in oriental lily cultivar ‘Sorbonne’ influences physiological and biochemical changes, resulting in shorter plants, smaller flowers, and shallower leaf color, affecting the quality of flowers [5]. The high temperature (>30 °C) reduces the quality of the cut flowers and leads to the degeneration of the bulb [6]. In addition, climatic variation has adverse effects on the ecological distribution of native Korean *Lilium* sp., leading to a steady decline in *Lilium* germplasm resources [7]. Therefore, it has become necessary to produce lily cultivars resistant to biotic and abiotic stresses to preserve lily resources world-wide.

RNA-Seq is a sequencing technique which uses next-generation sequencing (NGS) to reveal the presence and quantity of RNA in biological samples at a given moment, analyzing the continuously changing cellular transcriptome [8]. RNA-seq technology has been applied extensively in lily research to study differentially expressed genes (DEGs) in cold-stress response [9], flower color biosynthesis [10], and in vernalization [11]. Transcriptional regulation is the most useful link to study gene expression in plants. Transcription factors control various critical biological processes in gene transcription regulation networks [12,13]. The plant genome contains large number of genes acting as transcription factors, and at least 58 transcription factor families have been identified [14]. Approximately 8% of the genes in *Arabidopsis* and 4% in rice (*Oryza sativa*) have been identified as transcription factors [15,16].

The *WRKY* family, named after the highly conserved *WRKY* domain, is the largest transcription factor family [17]. The *WRKY* domain is associated with approximately 60 amino acids and comprises the most conserved short peptide sequence, *WRKY*GQK, with an adjacent C_2_H_2_ or C_2_HC zinc finger structure. *WRKY*GQK has various variants, such as *WRKY*GKK, *WRKY*DQK, and *WRKY*DHK [18,19]. Depending on the number of *WRKY* domains and zinc finger types, *WRKY* families are divided into groups I, II, and III. Group I has two *WRKY* domains with one C_2_H_2_-type zinc finger (C-X_4_–_5_-C-X_22_–_23_-H-X_1_-H), and group II has one *WRKY* domain with a C_2_H_2_-type zinc finger. Group II is further divided into five subgroups (IIa–e). Group III also has only one domain but with the C_2_HC-type zinc finger (C-X_7_-C-X_23_-H-X_1_-C)_6_. The *WRKY* gene family is one of the most crucial transcription factor families playing important roles in regulating developmental and physiological processes in plants [20]. *SPF1* was the first *WRKY* gene reported from sweet potato (*Ipomoea batatas*) in 1994 [21]. Subsequently, several *WRKY* genes have been identified in parsley (*Petroselinum crispum*) [22], *Arabidopsis* [23], rice [24], in *Lilium regale* [25], *Lilium* oriental hybrid ’Sorborne’ [26], *Lilium pumilum* [27], and *Lilium longiflorum* [28].

Along with other plants, the vegetatively propagating bulbing lily has a number of *WRKY* genes, and their organization and function are largely unknown. This study aimed to:
Systematically investigate *WRKY* genes in *Lilium* using the available transcriptome;Study the gene composition, construction of an orthologous dataset, and protein structure of WRKY;Explain the classification and evolutionary relationships among *WRKY* genes;Study the expression profiles of *LlWRKY* genes in different developmental stages, tissues, and biotic and abiotic stress conditions to identify suitable candidate genes for further functional analysis.

This work provides novel insights into evolutionary relationships, protein structures, and the expression profile of *LIWRKY*s, which will help the further investigation of *LlWRKY* genes.

## 2. Results and Discussion

### 2.1. In Silico Identification of the WRKY Gene Family

The *WRKY* gene family, as important transcription factors, plays a pivotal role in the regulation of plant development, growth, and stress response [29]. The functions of several *WRKY* genes in *Arabidopsis* and other model crops have been systematically studied [30,31,32,33]. To explore the function and organization of *LlWRKY* genes in *Lilium*, transcriptome data were generated and used to identify *WRKY* genes through computational analysis. Using a BLAST search, a total of 38 unique transcripts with high similarity was found. These *Lilium* putative transcripts were analyzed using the NCBI-CDD web server to obtain conserved protein domain information on sequences. Moreover, sequences with complete *WRKY* domains were further analyzed via BLAST in NCBI to remove repeats. Finally, 38 unique *LlWRKY* gene sequences were identified including 24 full CDS (Appendix A) and 14 partial CDS (Appendix A), and detailed information of the *LlWRKY* domains is substantiated in Appendix A.

### 2.2. Multiple Sequence Alignment, Protein Structure Prediction, and Conserved Motifs Analysis

The protein sequence-structure analysis of *LlWRKYs* showed that 33 *LlWRKYs* usually contained the highly conserved sequence ‘*WRKY*GQK.’ *LlWRKY*26, *-34*, and *-35* have an incomplete ‘*WRKY*GQK’ sequence, whereas *LlWRKY27* and *LlWRKY37* have an incomplete zinc finger domain. *LlWRKY29* and *-30* are deficient in the WRKY domain with a complete zinc finger domain in our study (Figure 1 and Appendix A). Two *LlWRKYs*, *LlWRKY*15 and *LlWRKY*16, were the most common sequence variants of ‘*WRKY*GKK’ (Figure 1). Seven *LlWRKY* protein sequences varied in single amino acids in their *WRKY* domain. Sequences with such domains are widely distributed in several species [34,35]. Some *WRKY* domains have shown unusual variations, e.g., five unusual variations in grape (*Vitis vinifera*) *VvWRKY* and seven variations in wheat (*Triticum aestivum*) *TaWRKY* [19]. The *WRKY* domain is the most vital structure in *WRKY* proteins. The *WRKY* domain with the *WRKY*GQK domain interacts with the TTGACY core motif to activate downstream genes [17]. The variants in the *WRKY*GQK domain could influence the normal activity of DNA binding [36]. *WRKY* genes lacking the *WRKY*GQK domain recognize binding sequences excluding W-box elements. Tobacco (*Nicotiana tabacuom*) *NtWRKY12* and soybean (*Glycine max*) *GmWRKY*6 proteins cannot bind to the W-box element but bind to WK-box (TTTTCCAC) [33]. In addition, variations can occur in the zinc finger domain of *WRKY* sequences. Three variations have been reported in three *VvWRKY* proteins. However, the function of the variants in the zinc finger domain is unknown, possibly influencing *WRKY* gene classification [19].

The MEME program predicted conserved motifs in *LlWRKY* proteins. The six identified motifs demonstrated *WRKY* protein structure in *Lilium* (Figure 2B). The details of six conserved motifs are shown in Appendix A and the conserved sequences of motif 1 and motif 2 in Appendix A. The number of motifs in *LlWRKY* protein ranged from 1 to 6, and the length of motifs ranged from 6 to 50 amino acids. Additionally, six motifs, namely motifs 1, 2, 3, 4, 5, and 6, were found in the *WRKY* domain. Motif 5 was located outside the *WRKY* domain (Appendix A). Meanwhile, motif 1 was shared by *LlWRKY*4, *-27*, *-37*, and *-38*, whereas motif 2 was shared by five *LlWRKYs*: *LlWRKY*26, *-29*, *-30*, *-34*, *-35*. Ten *LlWRKYs*: *LlWRKY*2, *-8*, *-11*, *-12*, *-13*, *-14*, *-17*, *-18*, *-22*, and *-25* shared motifs 1 and 2. Furthermore, *LlWRKY*19, *-20*, and *-21* from group III all shared motifs 5, 3, 1, and 2. Motifs 6, 3, 1, and 2 were shared by *LlWRKY*5, *-6*, *-9*, and *-10* from group IC. 

### 2.3. Classification and Phylogenetic Analysis

The *WRKY* gene family is highly conserved in both dicots and monocots. Based on sequence similarity and protein structure, it is divided into three groups [20]. To categorize *WRKY* sequences and investigate evolutionary relationships in *LlWRKY* proteins, 38 putative *Ll*WRKY proteins from *Lilium*, seven WRKY proteins from *Arabidopsis*, and 19 *WRKY* proteins from rice were selected for phylogenetic analysis. The phylogenetic analysis results in Figure 2A show that these *Ll*WRKY proteins are categorized into Groups I, II, and III. Then, we analyzed the evolutionary relationship among *LlWRKY* and WRKYs of different plants, including *Arabidopsis*, rice, *L. regale*, *L. hybrid* div VII, and *L. pumilum*. The phylogenetic results showed that there exists close relationship of *LlWRKY* with *WRKYs* of *L. regale*, *L. hybrid* div VII, and *L. pumilum* (Appendix A).

Seven *Ll*WRKY proteins constituted Group I, including sequences with two *WRKY* domains and one zinc finger motif, C_2_H_2_ (C-X_4_-C-X_22_-_23_-HXH). The 22 *Ll*WRKY proteins contained one WRKY domain and only one zinc-binding motif, C_2_H_2_ (C-X_4_-_5_-C-X_23_-HXH), constituting group II. The remaining nine *Ll*WRKY proteins were assigned to group III, with a single *WRKY* domain and C_2_CH (C-X_7_-C-X_23_-HXC) zinc-binding motif. According to the *WRKY* subgroup classification in rice and *Arabidopsis*, group II of *LlWRKYs* is further divided into five subgroups: group IIa (5), IIb (2), IIc (6), IId (3), and IIe (6).

### 2.4. Functional Annotation of LlWRKY Based on the Orthologous Gene in Lilium and Rice 

There are two types of homologous genes, orthologs and paralogs. Ortholog genes have diverged through speciation in different species, whereas paralog genes are duplicated in a single species. Orthologs are widely distributed in species and are assumed to elicit equivalent biological functions sharing key properties with other species [37]. Various methods are available to identify orthologous genes to study their function [38]. Phylogenetic analysis is a rapid, simple, and relatively accurate approach to evaluate orthologs widely present in different organisms.

Several orthologous genes have been identified in *L. longiflorum*- *LlWRKYs* (Appendix A). Among them, one *LlWRKY7* gene has only one homolog gene—*OsWRKY*36 in rice. Other *Lilium* genes have several homologous *OsWRKY* genes in rice. *LlWRKY14, -17, -19, -20, -21, -31, -32, -33*, and *-38* are homologous genes to *OsWRKY19, -22, -55*, and *-74* (Figure 3). At the same time, several *Lilium LlWRKY* genes have similar homologous genes in rice. *LlWRKY1, -2, -18, -23, -24, -26, -34, -35, -36,* and -*37* have the same homologous genes: *OsWRKY13,* -*14, -17, -39,* and -*68* in rice (Figure 3). Gene function can be putatively predicted according to their homology, as the functions of several *OsWRKY*s have been extensively studied [22,29].

In *Oryza sativa*, transgenic lines deficient in *OsWRKY-36* and *OsWRKY-102* genes and double-mutant lines OsWRKY36/OsWRKY102 significantly increased the lignin content. They are associated with the repression of rice lignification [39]. Over-expression of the stress-induced *OsWRKY45* enhances disease resistance and drought tolerance in *Arabidopsis* [32]. *PheWRKY1*, homologous of *OsWRKY10*, a gene from moso bamboo (*Phyllostachys edulis*), enhances disease resistance in transgenic *Arabidopsis thaliana* [40]. Overexpression of *LrWRKY4* gene significantly increased root-length and enhanced resistance against *B. cinerea* in *Arabidopsis* [25]. Similarly, overexpression of *LlWRKY39* gene in *L. longiflorum* seems to be related to protect cells from high temperature [28]. The homologous *OsWRKY* proteins in rice have been used as a reference to explore the potential roles of *LlWRKYs*. Based on functional annotation of rice homolog genes, four groups of *LlWRKYs* were identified (Figure 3). Of these, nine *LlWRKY* genes were involved in abiotic stress responses. In addition, 22 *LlWRKYs* were found to be involved in biotic stress responses, and the remaining seven *LlWRKYs* in other stresses and other functions.

Several *WRKY* genes have been studied for their response to abiotic and biotic stress [41,42]. In *Lilium regale*, 23 *LrWRKY* genes with complete *WRKY* domains have been identified. Six *LrWRKY* genes was induced by *B. cinerea* infection. *LrWRKY4*, *LrWRKY8,* and *LrWRKY10* were expressed to higher levels, while *LrWRKY6* and *LrWRKY12* were expressed to lower levels. *LrWRKY4* and *LrWRKY12* genes were responsive to salicylic acid (SA) and methyl jasmonate (MeJA) treatments [25].

Approximately 51% of *AtWRKY* genes from *Arabidopsis* in roots showed high expression under salt stress [43], and 57% of *OsWRKY* were differentially expressed under drought, cold, and salt stress [44], whereas 70–90% of *VvWRKY* genes were differentially expressed under various abiotic or biotic stresses [19].

### 2.5. Expression Analysis of LlWRKY Genes under Biotic Stresses in Lilium Longiflorum

Nine genes, *LlWRKY3, -5, -6, -9, -10, -11, -12, -13,* and *-20*, are associated with biotic stresses and one, *LlWRKY4*, related to another function were selected to determine the expression profiles in vivo (in field) and in vitro (laboratory) after *B. elliptica* infection. As shown in Figure 4A,B, the selected *LlWRKY* genes were induced to biotic stresses. All *LlWRKY* genes showed higher expression than uninfected *L. longiflorum*. In particular, *LlWRKY3, -4*, *-5*, *-10*, and *-12* had high expression levels in response to *B. elliptica* infection under both conditions—in vivo and in vitro. *LlWRKY4* showed a significantly high expression after *B. elliptica* infection in vivo and in vitro (Figure 4A,B). In in vivo analysis, the expression levels of five *LlWRKY* genes in infected leaf highly increased in comparison to un-infected leaf. The gene *LlWRK4* shows the highest expression—an increase by 152.287-fold. Similarly, *LlWRKY10*, *-5*, and *-3* showed a 36.286-, 36.062-, and 17.296-fold increase, respectively. In in vitro analysis, *LlWRKY*4 gene shows the highest expression, 30.1789-fold. Other genes showed expression in the following order: *LlWRKY10* > *LlWRKY5* > *LlWRKY12* > *LlWRKY3*. In in vivo and in vitro condition both, LlWRKY6, *-9*, *-11*, *-13*, and *-20* showed slight expression. Most of the *LlWRKY* genes from *L. longiflorum* have more expression in the in vivo condition than expression in the in vitro condition because of the natural environments. Plants in their natural habitat are constantly subjected to multiple fluctuating environmental factors, and complex genetic regulatory networks are functioning in them.

Furthermore, it has been found that orthologous genes *OsWRKY4* and *OsWRKY30* were related to group I. *OsWRKY30* gene overexpression increased the resistance to *M. oryzae* and *R. solani*, possibly activating various downstream genes, including jasmonic acid (JA) biosynthesis pathway genes [45]. *OsWRKY4* acts as a transcriptional activator and responds to defense against *R. solani* [46]. *LlWRKY4,* associated with other functions during functional annotation, is highly responsive to *Botrytis* infection.

*OsWRKY31* gene overexpression enhances disease resistance, affects auxin response, and affects root growth in transgenic rice [47]. *GhWRKY15*, identified in cotton (*Gossypium hirsutum*), is involved in plant development and disease resistance [48]. After *Botrytis* infection, expression analyses helped screen *WRKY* genes that might be involved in natural resistance against *Botrytis* infection in *Lilium.*

### 2.6. Expression Profiles of LlWRKY Genes under Abiotic Stresses

*WRKY* genes were found to confer to abiotic stress resistance, as reported by Chen [49]. Based on the transcriptome data and their orthologous gene information under different abiotic stresses (Appendix A), we selected *LlWRKY* genes associated with abiotic or biotic stress to analyze their expression characteristics [50]. *LlWRKY4*, *-*7, *-8*, and *-18* genes exhibited higher responses to drought stress in 10-d samples, whereas *LlWRKY1*, *-*2, *-3*, *-20*, and *-22* were weakly expressed in 5-d-old samples. However, *LlWRKY3, -22*, and *-8* did not express well in 5- and 10-d samples (Appendix A). *LlWRKY1*, *-2*, and *-4* genes were upregulated in heat treatments, maintaining a high expression level at 6 and 12 h, whereas *LlWRKY7* was not sensitive to high-temperature treatment (Appendix A). Moreover, *LlWRKY1* and *LlWRKY7* were highly expressed under cold treatment by more than 100 times. However, the expression level of *LlWRKY1* and *LlWRKY*7 initially increased and reached a maximum after 24 h (Appendix A). The expression of *OsWRKY76* was induced by low temperature; *OsWRKY*76 overexpression improved tolerance to cold stress in rice plants [51]. Some *OsWRKY* genes are induced by drought, salt, heat stress, and cold [24]. *OsWRKY*69 specifically binds to ABL1, which regulates rice stress responses [52].

### 2.7. Tissue-Specific Expression Patterns of LlWRKY Genes

The expression levels of 10 *LlWRKY* genes were analyzed via qRT-PCR in leaf, bud, stem, flower, stigma, internode, and root of lily (Appendix A). Most of the *LlWRKY* genes clearly showed tissue-specific expression. *LlWRKY4*, *-5*, *-6*, *-9*, *-11, -12*, and *-13* had higher expression in buds, stems, and roots, whereas *LlWRKY-10* and *LlWRKY-20* showed relatively lower expression levels in stems and roots, respectively. Furthermore, some *LlWRKY* genes did not show significant differences in expression levels. Additionally, some *LlWRKY* genes showed consistent levels of tissue-specific expression patterns and relatively strong expression in buds, stems, and roots.

*DcWRKY3*, *-8*, *-24*, *-64*, and *-88* from carrot (*Daucus carota)* are involved in root development, while *DcWRKY3* and *DcWRKY8* also play an important role in plant development [53]. Similarly, *OsWRKY78* was reported to participate in regulating of stem elongation in rice [54]. In our study, *LlWRKY3*, *-4*, *-5*, *-6*, *-9*, and *-10* were highly expressed at 40 d of development stage, and *LlWRKY3*, *-4*, *-5*, *-6*, and -*20* at 120 d, while only *LlWRKY20* was highly expressed at 60 d (Appendix A). The similar results showed in carrot and *Arabidopsis*, in which expression patterns of different *WRKY* genes depended on plant development or leaf stages, which seems to be associated with different roles in intracellular signaling by specific DNA binding features [53]. Our result provided basic clues on *LlWRKY* genes playing functional roles during the growth and development in *Lilium*.

## 3. Materials and Methods

### 3.1. Plant Materials

All plant materials were obtained from Kangwon National University, Chuncheon, Gangwon-do, Korea. Plants with brown spots or fuzzy, grayish mold on leaves, buds, and flower petals were selected first, and whole plants were collected at 8:00 am from fields of the University. Five leaves from each plant were taken for the experiments. Samples were kept in a −80 °C freezer until liquid nitrogen was used. RNA was isolated from approximately 1.0 g of leaf tissue using a Hybrid-R™ Total RNA isolation kit (GeneAll, Daejeon, Korea). The quantity and quality of RNAs were determined using a NanoDrop Lite spectrophotometer (Thermo Fisher Scientific, Waltham, MA, USA). cDNA synthesis was conducted using the TruSeq mRNA sample preparation kit. Sequencing libraries were prepared according to the manufacturer’s instructions (Illumina) for cluster analysis and sequenced using Illumina HiSeq™ 2500 sequencing technology (Illumina). Illumina paired-end reads were trimmed using the Dynamic Trim of SolexaQA software package. Transcripts were assembled using Trinity. In the gene annotation process, gene expression values were normalized using the DESeq software package. The assembled total sequences of *Lilium* were annotated in the NCBI database using BLASTX.

### 3.2. Identification of WRKY Gene Family in L. longiflorum

Transcriptomic data from Illumina sequencing reads were deposited at the National Agricultural Biotechnology Information Center (NABIC) (accession: **NN-3727-000001** and **3728-000001**). A total of 38 *WRKY* gene sequences of *L. longiflorum* were obtained from our RNA sequence data above (unpublished). For the transcription factor (TF) annotation, the TF database (PlantTFDB) version 4.0 was used and classified into families. *WRKY* gene sequences were queried against the PFAM (http://pfam.xfam.org/search accessed on 15 February 2021) and CDD (http://www.ncbi.nlm.nih.gov/Structure/cdd/wrpsb.cgi accessed on 15 February 2021) databases to search for their putative protein domain features. Open reading frames (ORFs) of *WRKY* gene sequences were obtained using the NCBI’s ORF finder. Subcellular localization of *WRKY* proteins was predicted using TargetP 1.1 and SignalP 4.1 [55]. Theoretical pI/molecular weight of proteins was obtained using the Compute pI/Mw tool on the ExPASy platform (http://web.expasy.org/compute_pi/ accessed on 15 February 2021).

### 3.3. Multiple Sequence Alignment, Protein Structure Prediction, and Conserved Motifs Analysis

*Arabidopsis thaliana* and rice *WRKY* genes were obtained from previous studies [56]. The evolutionary relationships among these *WRKY*s were investigated by aligning predicted amino acid sequences of *WRKY* genes from *L. longiflorum* with the *WRKY* genes of other species using the ClustalX program, and sequence identity and similarity were obtained. *LlWRKY* gene families were obtained using the MEME tool (http://meme.nbcr.net/ 15 February 2021) [57] with the following parameters: 0 or 1 motifs per sequence, minimum amino acid motif size of 6 and maximum of 50, and 2–6 optimum amino acid sites for each motif. 

### 3.4. Phylogenetic Analysis

Using a dataset of 64 predicted amino acid sequences of the *WRKY* gene family members from *Lilium* and different plant species, sequence alignment was conducted using MEGA7 [58] of the MUSCLE program [59] with the following parameters: maximum iterations 1000; hydrophobicity 1.2; gap open 1. MUSCLE align sequences were based on multiple sequence comparison by log-expectation. 

For phylogenetic analysis, the alignments made above were imported into MEGA7. A maximum-likelihood tree, based on the Jones-Taylor-Thornton model was built with a pairwise deletion option and 1000 bootstrap replicates, and the evolutionary relationships among *LlWRKY* gene family members were determined.

### 3.5. Abiotic and Biotic Stress Treatments

Abiotic stress treatment. Selected Bulbs of *Lilium longiflorum* were planted in 9-cm pots filled with soil (*v*/*v*/*v*, 1:1:1 = sterile peat: vermiculite: perlite) at 24/16 °C temperatures in a day/night cycle with a 16/8 h photoperiod in a growth chamber for six weeks. 

For cold and hot stress, 4 °C growth chamber for cold treatment and 40 °C growth chamber for heat treatment was separately maintained before use. Six-week-old plants were kept in a cold chamber and a hot chamber for 24 h. Leaf samples were collected at 0, 1, 6, 12, and 24 h during stress treatment. These samples were immediately kept into liquid nitrogen and stored at −80 °C for RNA isolation. For drought stress, six-week-old plants were kept in a growth chamber maintained at 24/16 °C temperatures in day/night cycle, with 65% relative humidity, and with a 16/8 h photoperiod. During drought treatments, control plants were watered every 2 d, treatment plants were watered every 5 d and 10 d for mild and severe drought stress treatments, respectively. The soil moisture content was measured before and after drought treatment using SM150 Moisture meter (Delta-T Devices, Cambridge, Great Britain). At the beginning of drought stress, the soil water content was recorded 30% for control, 10% on the 5th d, and 3% on the 10th d during the stress period. Leaf samples were collected from the control and drought treated plants- the 5th d and the 10th d; then, samples were immediately frozen in liquid nitrogen and stored at –80 °C. 

Biotic stress treatment. Scales of the bulb of *Lilium longiflorum* were washed with tap water, then sterilized by ethanol 70% (*v*/*v*) for 30 s, rinsed five times with distilled water, and placed in sodium hypochlorite 10% for 5 min, then washed five times with sterilized distilled water. Scales were cultured in MS medium [60], supplemented with 0.1% myo-inositol, 3% sucrose, and 0.7% agar, for three weeks for the proliferation of blub. Then, scales were transferred to MS medium, supplemented with 3% sucrose, 0.1% myo-inositol, and 0.7% agar, for 3 weeks to obtain leaves. Cultures were placed in incubation room at the 24 ± 2 °C under 16/8 h light (a fluence rate of 240 µmol·m^−2^·s^−1^)/dark cycles during all stage.

*B. elliptica* (KACC43461) isolate was grown on petri dish containing 8% potato dextrose agar (PDA) (Becton, Dickinson and Co., USA) at 25 °C under UV light 12/12 h light/dark cycle for one weeks before use. Then, sterile distilled water (10 mL) was added to suspend the mycelium. Spore suspension were stained with methylene blue and spore-density was measured by a hemocytometer on an optical microscope (Nikon). The concentration of spore suspensions was adjusted to reach final concentrations of 2.0 × 10^4^ spores·mL^−1^.

The detached leaves from 3-week-old *Lilium* plant were washed three times with sterile distilled water, after which leaves were placed in a cell culture plate containing spore suspension of *B. elliptica* spores (2.0 × 10^4^ spores·mL^−1^) and inoculated at 25 °C for 12 h in the dark. Then, leaves were washed with 70% ethanol after inoculation and cultured on 0.7% agar plates in a 16/8 h light (fluency density of 130 μmol m^−2^·s^−1^ at 25 °C. Sterile distilled water was used as a control for *B. elliptica* treatment. Disease symptoms were detected and assessed at 3, 5, and 7 d after inoculation (DAI). The disease index was measured by the percentage of lesion areas compared to control leaves, which were placed in sterile distilled water. The disease index (DI) was divided into 0–4 according to how much leaf area is infected; 0 = no lesions observed, 1 = 1–10%, 2 = 11–25%, 3 = 26–50%, and 4 = over 50% of the leaf area infected [61]. Total RNA was extracted from both uninfected and infected leaves for qRT-PCR analysis.

### 3.6. RNA Isolation and qRT-PCR

Total mRNAs were isolated from normal and differently stressed treated tissues including mature and young leaves, stems, and roots using a Hybrid-R™ Total RNA isolation kit (GeneAll, Daejeon, Korea), as described in the manufacturer’s instructions. cDNA was synthesized from extracted total RNA using PrimeScript^®^ RT reagent with gDNA eraser kit (Takara Korea Biomedical, Seoul, Korea). Gene-specific qRT-PCR primer pairs for all 16 *LlWRKY* genes were designed using the Primer3plus web tool (http://www.bioinformatics.nl/cgi-bin/primer3plus/primer3plus.cgi accessed on 15 February 2021) (Appendix A). A 20-μL reaction volume was prepared containing 2 μL of reverse-transcribed cDNA, SYBR^®^ Green mix (PhileKorea, Daejeon, Korea), and 0.5 μM of gene-specific forward and reverse primers. To estimate the relative RNA expression level, PCR reactions with parameters: one cycle at 95 °C for 3 min, 40 cycles of 95 °C for 15 s, 60 °C for 20 s, and 72 °C for 15 s, a cycle at 65 °C for 5 s, and a final cycle of 95 °C for 2 s, were performed with SYBR Green-based qRT-PCR using an Eco™ Real-Time PCR System (Illumina, Seoul, Republic of Korea) to detect primer specificity based on melt curve analysis and 1% agarose gel electrophoresis for the amplification product. For each gene, three biological replicates with three technical replicate PCR reactions were conducted. The expression levels of candidate genes were normalized against the housekeeping gene actin-like gene *LP59* from *Lilium longiflorum* accession DQ019459 [1] using the 2^−ΔΔCT^ method [62].

## 4. Conclusions

In this study, 38 *LlWRKY* genes were identified and studied to investigate the organization and abundance of *WRKY* in *Lilium*. These *LlWRKYs* were analyzed for protein structure, figuring out related motifs, genes classification according to their functions, and phylogenetic relationships by comparing with model plants. Furthermore, after a comparison with other plants, orthologous gene analysis predicted the biological function of *LlWRKYs*. Finally, qRT-PCR analysis using transcriptional profiles was performed in various tissues under biotic and abiotic stresses, demonstrating tissue-specific and stress-responsive *LlWRKY*s. This study provides helpful information for further investigation of the function of *LlWRKY* in lily as stress-tolerant genes.

## Figures and Tables

**Figure 1 plants-10-00776-f001:**
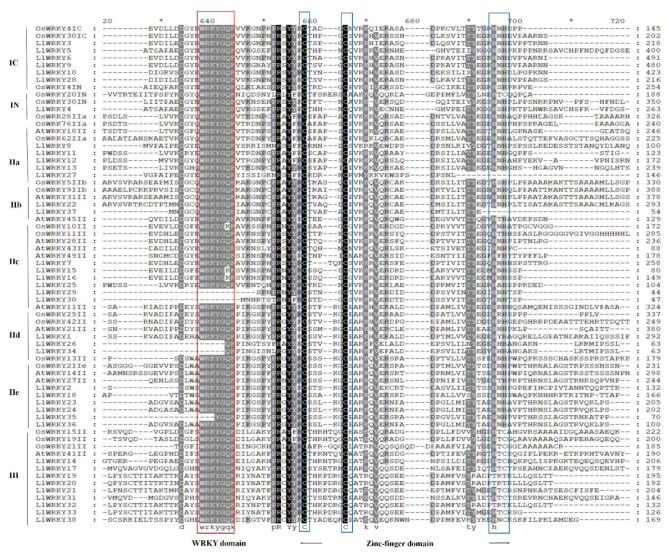
Multiple sequence alignment of various types of *WRKY* domains (I, II, and III) from *Lillium* (*LlWRKY*), *Arabidopsis* (*AtWRKY*), and rice (*OsWRKY*). Identity, similarity, and deletions among *WRKY* alignments are indicated by black and grey shades and hyphens, respectively. A red box represents the WRKY domain and a blue box the zinc finger domain. N means N-terminal and C means C-terminal.

**Figure 2 plants-10-00776-f002:**
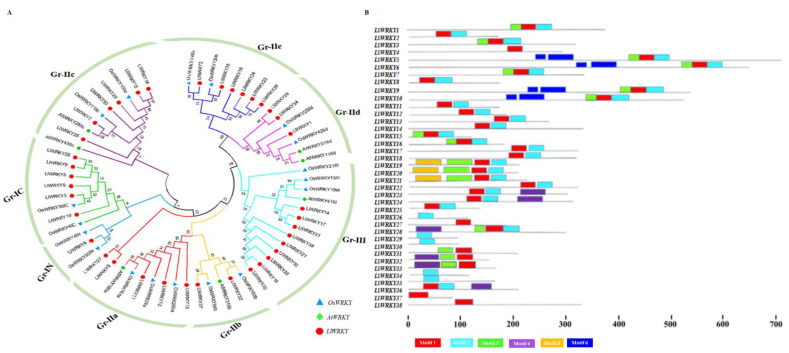
Unrooted phylogenetic tree analysis of *WRKY* proteins: (**A**) *Lilium* (38), rice (19), and *Arabidopsis* (7); (**B**) conserved motifs of 38 *LlWRKY* proteins. The phylogenetic tree was constructed using MEGA7. Different colors represent different groups and red circles represent LlWRKY, blue triangles represent *OsWRKY* and green squares represent *AtWRKY* proteins. MEME was used to predict motifs, and boxes represent these motifs.

**Figure 3 plants-10-00776-f003:**
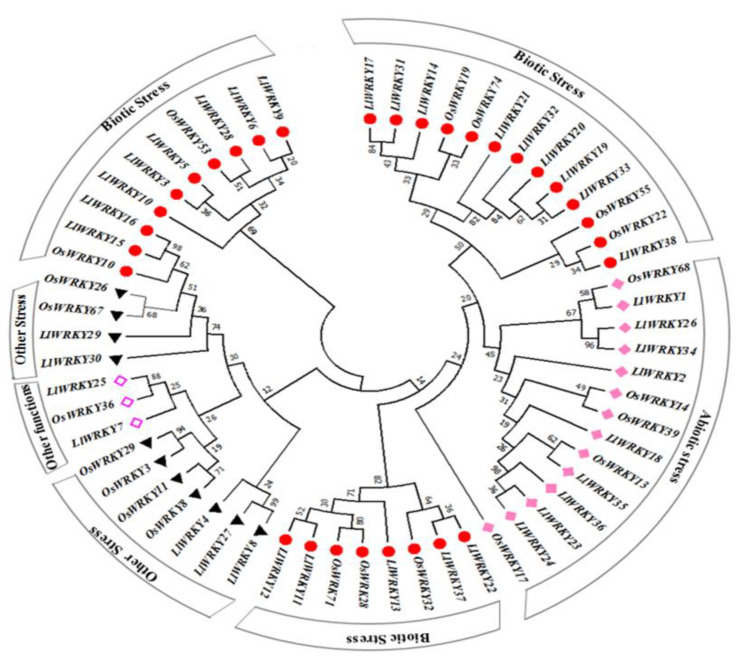
Functional grouping of the annotated *LlWRKY* proteins based on the homologous gene information regarding biotic stress-responsive genes. The unrooted phylogenetic tree was constructed for *Lilium longiflorum* and rice proteins to identify homologous gene pairs. Red circles represent the responsive genes to biotic stress; pink squares, black triangles, and magenta squares represent genes responsive to abiotic stress-responsive, other stresses, or possessing other functions, respectively.

**Figure 4 plants-10-00776-f004:**
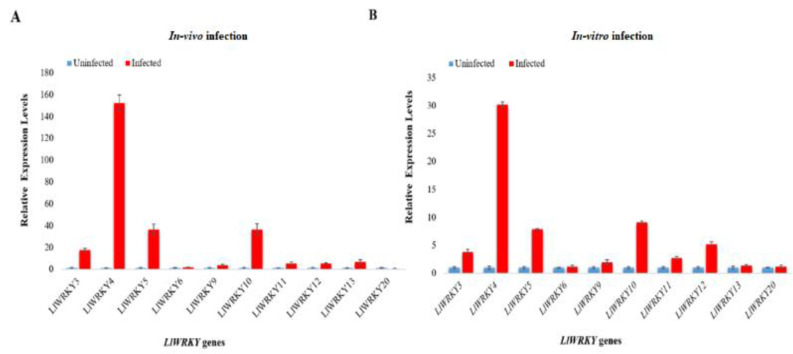
Comparative expression patterns of the selected *LlWRKY* genes in response to *Botrytis elliptica* infection investigated under (**A**) in vivo and (**B**) in vitro conditions. Error bars indicate the standard error of three biological replicates and the expression levels were normalized against the level of *actin*.

## Data Availability

The data are available in the article and Appendix A.

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
