# Peer review of "Genome-Wide Transcriptomic Identification and Functional Insight of Lily WRKY Genes Responding to Botrytis Fungal Disease"

_plants, 2021, doi:10.3390/plants10040776_

Round 1

Reviewer 1 Report

This work provides novel insights into evolutionary relationships, protein structures, and the expression profile of LIWRKY genes for use in the genetic engineering in Lilium longiflorum against Botrytis disease.

This article presents interesting information, however, I have some points that require clarification or rewritting, and I hope these commenrs may be helpful for authors to improve this manuscript.

Introduction section

In the introduction part, the authors should provide more information about effects of Botrytis diseases on Lily.

In addition, the introduction should include more information regarding previous works about the analysis of WRKY transcription factors and characterization of Botrytis -responsive LrWRKY genes from Lilium

Results and discusion section:

Lines 166-174. this part of discussion is poor, and the authors just listed references, but didn’t discuss their data with previous studies well. I also recommend including information about the Botrytis -responsive LrWRKY genes from previous works in Lilium.

Overall, in the discussion part, the authors should discuss more about the biological significances of their data in Lilium.

Material and Methods section:

Lines 243-248. The authors conducted the experiment under in vivo and in vitro conditions. The authors did not mention how the Botrytis elliptica infection was carried out, and lack of the necessary information about both conditions. Please add the related information.

Lines 299-314, where is the total RNA for qRT-PCR validation from? How many replicates were used for qRT-PCR validation? More details needed here.

Lines 313-314. Authors indicate that the expression levels of candidate genes were normalized against the house- keeping gene actin using the 2-ΔΔCT method. Did the authors do amplification efficiencies analysis before using the 2−ΔΔCt method?

Author Response

Dear Sir

Reviewer 2 Report

The manuscript, “ Genome-wide transcriptomic identification and functional insight of lily WRKY genes responding to Botrytis fungal disease,” presented by Shipra Kumari and colleagues, is a research paper on the identification and the structural organization of WRKY genes from Lilium longiflorum. Using bioinformatic resources, the authors presented a classification of putative LlWRKY proteins according to their structural features. Furthermore, in this work, the experimental results of transcriptome analysis of LlWRKY genes in different organs of lily and expression profiling of these genes in leaves under the abiotic and biotic stresses are presented.

Simultaneously, the article has significant flaws that reduce its scientific importance and do not allow recommending it for publication.

I can't entirely agree with the authors that ”the vegetatively propagating bulbing lily has limited WRKY genes, and their organization and function are completely unknown” (lane 68-69). In 2018 Q. Cui and colleagues reported the identification of 23 WRKY genes in Lilium regale (Q. Cui, X. Yan, X. Gao, D.-m. Zhang, H.-b. He, G.-x. Jia, Analysis of WRKY transcription factors and characterization of two Botrytis cinerea-responsive LrWRKY genes from Lilium regale, Plant Physiology et Biochemistry (2018), doi: 10.1016/j.plaphy.2018.04.027).

The author wrote that “several WRKY genes have been identified in parsley (Petroselinum crispum) [16], Arabidopsis [17], rice [18],” but WRKY families in these species consist of several dozen genes.

Line 73, “protein structure of WRKY genes,” is this confusing?

Line 118-119. The authors point out that “The seven identified motifs demonstrated WRKY protein structure in Lilium” but figures 2B and S1 were shown only 6.

Further, line 122-123. “Additionally, six motifs, namely motifs 1, 2, 3, 4, 5, and 6, were found in the WRKY domain. Motif 5 was located outside the WRKY domain (Fig. S3).” But Figure S3 is “Comparison of expression profiles of the selected LlWRKY genes under different abiotic stresses….”

The authors did not describe the principle of choosing WRKY proteins from Arabidopsis and rice for phylogenetic analysis.

Line 188-189. The author wrote that “All LlWRKY genes showed higher expression than uninfected L. longiflorum”. But this is not obvious in Figures 4A, B. The authors did not explain why the differences in the transcriptional profiles of plants infected in vitro and in vivo.

The reference to Table S4 (line 216), which shows the sequences of primers, looks incorrect.

It is not clear why infected plants were chosen to study gene expression in different organs. Are these genes not expressed in uninfected plants?

The quality of the figures is low, the designations on them are hardly distinguishable, the figures legends are not informative.

All mentioned above comments are just a small list of insufficiencies that do not allow me to recommend the manuscript for publication.

Author Response

Dear Sir/ Madam

Reviewer 3 Report

The study by Kumari et al. is a comprehensive bioinformatic analysis of the WRKY gene family in Lillium sp. It also includes transcriptomic data designed to identify possible functions of these stress- and development-related transcription factors (TFs). Although it is predominantly theoretical in nature, the study is well-designed and properly described and represents a welcome contribution to the knowledge of these important family of TFs. However, there a number of elements of this study that must be addressed before it is ready for publication in Plants. These are the following:

Major concerns:

  1. Lines 67-70: At the end of the “Introduction” the authors claim that “…the vegetatively propagating bulbing lily has limited 68 WRKY genes, and their organization and function are completely unknown”. This is a misleading statement. There at least four other recent study that have analyzed the WRKY TFs in lily plants from different perspectives. The authors are encouraged to include this relevant information in their study, which is contained in the following reports: Du et al. (2017) Identification of differentially expressed genes in flower, leaf and bulb scale of Lilium oriental hybrid ‘Sorbonne’ and putative control network for scent genes BMC Genomics 18:899; Cui et al. (2018) Analysis of WRKY transcription factors and characterization of two Botrytis cinerea-responsive LrWRKY genes from Lilium regale. Plant Physiology and Biochemistry 127:525-536; Wang et al. (2018) Transcriptome profiling provides insights into dormancy release during cold storage of Lilium pumilum. BMC Genomics 19:196; Ding et al. (2021) LlWRKY39 is involved in thermotolerance by activating LlMBF1c and interacting with LlCaM3 in lily (Lilium longiflorum). Horticulture Research 8:36. In addition, the mild statements included at the end of this section are not in accordance with the justification of this study provided at the beginning, where it is stressed that a better knowledge of these genes could help ameliorate the decline of lily flower production and the decline of Korean Lilium germplasm resources caused by increased biotic-stress pressure caused by global warming.

2.In Figure 1, please indicate in the legend the identity of the WRKY sequences shown in this figure: Lillium (LiWRKY), Arabidopsis (AtWRKY) and rice (OsWRKY), respectively. Additionally, please indicate the meaning of the blue boxes enclosing certain amino acid sequence segments.

  1. In Figure 2, please improve significantly the resolution of the images contained in this figure. They are unintelligible in their present form. Although the resolution of Figure 3 is higher, it must be nevertheless improved. Several changes to the legend of Figure 3 should be made as well: change "groping" to "grouping"; change “…based on the homologous gene information” to “…based on homologous gene information regarding biotic stress-responsive genes" and change to “pink squares, black triangles and magenta squares represent genes responsive to abiotic stress-responsive, other stresses or possessing other functions, respectively”.
  2. In Figure 4: suggest the change from “Relative Quantitation” to “Relative Expression Levels”. In the respective figure legend, please mention briefly how were the genes quantified and which Lillium genes were used for normalization. The number of plants used for each assay should be included too.
  3. In “Materials and Methods”, the authors should indicate how many genotypes were utilized in this study. This, considering their previous mention of the the menace that global warming represents to Lillium germplasm, at least in Korea. Is the brief mention of the field sampling procedure supposed to be the "in vivo" infection experiment? The description is lacking in essential details. For instance, a much more robust evidence, apart from visual inspection of the disease symptoms, is needed to determine that the plants were effectively infected by this particular pathogen. At what time of the day were the leaves sampled? How many leaves per plant were sampled? How many locations were included? How many different genotypes were sampled? How were the leaves manipulated before reaching the laboratory? Was there evidence other type of damage on the leaves, perhaps caused by mechanical injury, insect herbivory, water-stress, etc.? Please notice that no mention of the “in vitro” infection experiments is made in this section.
  4. The description of the “Abiotic and biotic stress treatments” is also deficient. Please define with greater precision the seasonal conditions in which the greenhouse experiments were performed. The description of a water-deficit experiments is very poor. The water content of the soil and the plants is required to determine the severity of the water-deficit stresses applied. How long were the plants exposed to low or high temperatures? Why were these particular temperatures selected for the experiments? How many leaves were infected? What was the infection severity chosen to collect the leaves? What were the disease symptoms? What are the defining characteristics of the pathogen used? Is it a laboratory strain? How was it isolated? How virulent is it?

Minor concerns:

  1. Line 11: In the “Abstract”, does the inclusion of the phrase “Genome-wide transcriptomic profiles of lily (Lilum longiflorum) have been expected to identify valuable genes” suggest that they have mostly failed to fulfill expectations?
  2. Line 22: In “Abstract”, please clarify the meaning of “molecular functions”.
  3. Lines 27-28: In “Abstract”, the conclusion appears to disregard the work that was performed in this study to determine additional roles of these TFs in abiotic stress responses and distribution in the plant.
  4. Lines 71-78: Please, start numbered statements with capital letters.
  5. Lines 118-128: Please re-write this passage to improve its readability.
  6. Line 152: Change "are diverged" to "have diverged".
  7. Line 156: Change "to evaluating" to "to evaluate".
  8. Lines 158-168: Please re-write this section of the MS. The message is repetitive.
  9. Line 188: Perhaps change “were sensitive” to “were induced/repressed/affected, etc.”
  10. Line 214: Change “WRKY genes responded to abiotic stress resistance…” to “WRKY genes were reported, reported to confer abiotic stress resistance...”
  11. lines 218 and 221: the terms “10-day-old samples” and “5-day old samples” make no sense in terms of water deficit stress. See observation, above.
  12. Lines 220 and 221: Please change “did not express well” to “were weakly expressed”.
  13. Please revise “References section for inconsistencies in the format. For example: Plant physiol., in line 372; 1870-4, in line 394 and in many others; BMC Plant Bio., in line 430, and Biochimica et Biophysica Acta, in lines 443-444.

Author Response

Dear Sir/Madam

Reviewer 4 Report

The article describes the characterization of the WRKY family of transcription factors in lily by bioinformatics and qRT-PCR. For the most part the paper reads well and the experimental design is appropriate, though not always well explained.

Major issues

The paper mentions transcriptomics, but do not describe ant transcriptomics in the results. EST sequencing is mentioned initially, but the methods describe an RNA-seq approach. I assume that authors want to publish another paper on RNA-seq, and don’t want to reveal their RNA-seq results here in detail; this is fine, but should be clearly stated.

ESTs are not the same thing as Illumine reads; were EST used in this study in addition to paired-end reads? This point needs to be clarified.

Importantly, more effort is needed on connecting experiments with the significance of this research.

qRT-PCR experiment revealed differential expression, though it is unclear if the only reference gene used is stably expressed under all conditions tested.

Additional points

Introduction

Page 2, line 33

“Lily (Lilium spp.) is one of the most important ornamental plants used for cut or potting flowers and gardening purposes.”

It would be nice to support this with some numbers.

Page 2, line 39

“Thus, the quality and yield of Lilium are affected by various abiotic and biotic stresses.”

Again, some estimates on the yield loss due to stresses, particularly abiotic stresses, would be helpful.

Page 2, line 44

“Expressed sequence tag (EST) analysis provides a basis for selecting resistance-related genes. ESTs are incomplete, randomly selected single-pass and unedited sequences obtained from complementary DNA libraries and are highly important for molecular analysis.”

I’m still confused if the transcriptomics done here were based on EST sequencing or RNA-seq. The methods describe RNA-seq, so why even talking about ESTs here? Clarify!

Page 2, 53-67

Summarize this paragraph; too much detail to keep readers’ interest. 3 groups can be mentioned, but no need to specify each sub-group.

Page 2, line 73

“study gene composition, ortholog construction, and protein structure of WRKY genes”

ortholog construction? How about: “construction of an orthologous data set”

Page 2, line 73

“explain the classification and evolutionary relationships among WRKY genes”

Did you explain the evolutionary relationship? I suggest to take out this claim

Page 4, line 122

“Additionally, six motifs, namely motifs 1, 2, 3, 4, 5, and 6, were found in the WRKY do-122 main. Motif 5 was located outside the WRKY domain (Fig. S3).”

Is Motif 5 found inside the WRKY domain, outside, or both? Also, Fig S 3 does not seem to have anything top do with this.

Page 5, Fig. 2

The text in this figure is not readable.

Page 5, line 155

“Phylogenetic analysis is a rapid, simple, and relatively accurate approach to evaluating orthologs widely present in different organisms.”

Vague, what is the approach you are using?

Page 6, Fig. 3

Add a legend to clearly show what the various symbols mean.

Page 6, line 170

“Four groups were identified”

Only 3 are mentioned; is four a typo?

Page 6, line 187

“in vivo and in vitro after B. elliptica infection”

Describe somewhere what exactly you mean here by “in vitro” and “in vivo”

Page 6, line 192

“Furthermore, it was found that the orthologous genes OsWRKY4 and OsWRKY30 were related to group I.”

These are rice genes; is this grouping a result from this study? If not, use “has been found” and cite the appropriate paper.

Page 6, line 199

"All 10 biotic stress-related LlWRKY genes".

Unclear to me, is it 9 or 10?

Page 6, line 201

LlWRKY12 does not show "much stronger" expression than some of the other genes after infection.

Page 7, Figure 4

If “Uninfected” is used as comparison to calculate fold change, it doesn’t need to be shown in the bar graph (it will just show up as “1” every time)

Page 7

The are 3 sections numbered 2.5.

Also, end each of the 2.5 section with some discussion of the results.

Page 8

“RNA was isolated from approximately 246 1.0 g of leaf tissue using TruSeq Stranded Total RNA sample preparation kits (Illumina,…)”

This is the first time that RNA-seq is mentioned, earlier ESTs are mentioned.

And this is NOT a kit to isolate RNA.

Page 8, line 273

“MEME tool (http://meme.nbcr.net/) [22] with the following parameters: 0 or 1 motifs per sequence”

Why restricting this search to motifs that are only found 0-1 times per sequence? There could be interesting motifs that occur more than once; why excluding those?

Page 9, 3.6. RNA isolation and qRT-PCR

More information would be helpful. What regions where targeted; how was specificity of primers to the various members of the same gene family ensured?

Was the amplification efficiency determined? If so, was it taken into account for quantification?

How was ensured that the only reference gene used was evenly expressed under all conditions?

Author Response

Dear Sir/Madam

Round 2

Reviewer 1 Report

All the comments have been taken into account and the manuscript can be accepted in present form

Author Response

 Thank you for review.

Reviewer 2 Report

Dear Authors,

I consider that the manuscript needs further major revision.

In section 2.1, it is necessary to substantiate including the short sequences with an incomplete domain in the number of unique genes: LlWRKY19, LlWRKY20, LlWRKY26, LlWRKY27, LlWRKY29, LlWRKY30, LlWRKY32, LlWRKY34, LlWRKY35 LlWRKY37 (Fig. 1, 2B, Table S1, and S2)

In section 2.2, the legend of Figure 1 is very confusing. The figure shows a comparative analysis of LlWRKY domains, not LlWRKY proteins. Use, please, the terms "protein", "domain", "motive" correctly. Share, please, the sequences of conservative motifs identified by the program MEME in proteins. For example, motif 1 contains a conservative sequence WRKYGQK, and motive 2 contains a conservative zinc-finger structure. That information will provide a comprehensive characterization of the structure of LlWRKY proteins.

In section 2.3, I would like to advise authors to read the WRKY domain structure description (lines 62-64) and not confuse it with the sequence (motif, consensus) WRKYGQK and correct this misunderstanding throughout the text. Why are proteins LlWRKY3, -4, -28 assigned to group 1 if they lack the second WRKY domain and proteins LlWRKY29, LlWRKY30 - to group 2, although they lack a WRKY motive? It is incorrect to use only the zinc-finger sequence for classification since it is characteristic of many transcription factors.

General remark for all sections, whose present gene transcriptional analysis results under the influence of stress factors, the change in gene expression under treatment can only be assessed compared to the control group that was not exposed. Therefore, all graphs should show the level of gene expression in control.

Lines 256-258, “5- and 10-days samples” what is it?

In section 2.7, discussing the role of WRKY proteins in the development of Lilium, the results of the expression of their genes in different organs of inoculated plants are presented (Figure S4). Why not in healthy plants?

The results presented in Figure S2 are not discussed in the manuscript.

Reviewer 3 Report

The revised version of MS by Kumari et al. titled “Genome-wide transcriptomic identification and functional insight of lily WRKY genes responding to Botrytis fungal disease” incorporated the necessary changes suggested to improve its quality and informative value. However, minor typographical errors were still detected. The authors should be able to easily correct them after a careful perusal of the manuscript. Moreover, the quality of the images is still not optimal: ¿Could their resolution be further increased? Once these minor corrections are implemented, the manuscript should be ready for publication in “Plants”.
